# Advances in Biomimetic Nerve Guidance Conduits for Peripheral Nerve Regeneration

**DOI:** 10.3390/nano13182528

**Published:** 2023-09-10

**Authors:** Faranak Mankavi, Rana Ibrahim, Hongjun Wang

**Affiliations:** Department of Biomedical Engineering, Semcer Center for Healthcare Innovation, Stevens Institute of Technology, Hoboken, NJ 07030, USA; fmankavi@stevens.edu (F.M.); ribrahim@stevens.edu (R.I.)

**Keywords:** nerve guidance conduits, regenerative medicine, biomimetic, peripheral nerve regeneration

## Abstract

Injuries to the peripheral nervous system are a common clinical issue, causing dysfunctions of the motor and sensory systems. Surgical interventions such as nerve autografting are necessary to repair damaged nerves. Even with autografting, i.e., the gold standard, malfunctioning and mismatches between the injured and donor nerves often lead to unwanted failure. Thus, there is an urgent need for a new intervention in clinical practice to achieve full functional recovery. Nerve guidance conduits (NGCs), providing physicochemical cues to guide neural regeneration, have great potential for the clinical regeneration of peripheral nerves. Typically, NGCs are tubular structures with various configurations to create a microenvironment that induces the oriented and accelerated growth of axons and promotes neuron cell migration and tissue maturation within the injured tissue. Once the native neural environment is better understood, ideal NGCs should maximally recapitulate those key physiological attributes for better neural regeneration. Indeed, NGC design has evolved from solely physical guidance to biochemical stimulation. NGC fabrication requires fundamental considerations of distinct nerve structures, the associated extracellular compositions (extracellular matrices, growth factors, and cytokines), cellular components, and advanced fabrication technologies that can mimic the structure and morphology of native extracellular matrices. Thus, this review mainly summarizes the recent advances in the state-of-the-art NGCs in terms of biomaterial innovations, structural design, and advanced fabrication technologies and provides an in-depth discussion of cellular responses (adhesion, spreading, and alignment) to such biomimetic cues for neural regeneration and repair.

## 1. Introduction

The nervous system is a complex network that acts as the command center to relay information via electrical impulses to control the movement, thoughts, senses, and automatic reflexes of the body. It is organized into the central nervous system (CNS), comprising the brain and spinal cord, and the peripheral nervous system (PNS), containing all the nerves and ganglia that connect the CNS to the body [1]. 

Peripheral nerve injury (PNI) is characterized by disruption of the information relays between the CNS and the body caused by traumas leading to neural cell damage or death. These injuries commonly occur for 2–5% of patients worldwide who experience physically traumatic events such as car accidents, joint dislocation, or surgery [2]. PNIs can be categorized into three main types: neurapraxia, axonotmesis, and neurotmesis. Neurapraxia refers to a temporary loss of function without disruption in nerve continuity. Axonotmesis injuries completely sever the nerve axon and the surrounding myelin, leading to functional loss. Neurotmesis causes a complete loss of function due to the disconnection of a nerve. To functionally regenerate the damaged nerves, it is necessary for axons to regrow and to be remyelinated via Schwann cells (SCs) [3].

Various strategies have been exploited to assist in or facilitate the establishment of a connection in recognition of the limited regeneration capacity of neural cells. Surgical interventions are commonly utilized as the primary approach to treat PNIs. Specifically, the prevailing standard procedure for small nerve gaps of less than 1 cm involves neurorrhaphy, which can effectively restore sensation and motor function to the peripheral nerve [1,4]. Large nerve gaps between 1 and 3 cm can also be fully regenerated with interventions such as nerve grafting or commercial nerve guidance conduits (NGCs) [5]. However, substantial nerve gaps exceeding 3 cm, known as critical nerve injuries [6,7], necessitate grafts to bridge the gap. Conventional treatments typically rely on autologous grafts of sensory nerves or decellularized nerve allografts. Clearly, autologous nerve grafting is the gold standard [4]. However, it faces foreseeable challenges, such as an inadequate source of donor grafts, the loss of nerve function at the donor region, a mismatch between donor and damaged nerves, and the potential formation of painful neuroma [8]. On the other hand, allografts are also confronted with several noticeable limitations, such as mismatched size and geometry with the recipient, unreliable supply sources, and possible immune rejections [9].

To address the above-mentioned limitations, efforts have been made to inject cells and biomolecules directly into the injury site following PNIs in anticipation of functional regeneration [10]. As noted, this strategy has yielded some promising outcomes in animal models; however, its clinical translation would be limited considering the difficulties in choosing the optimal cell types for transplantation and achieving cell confinement and survival at the injury site without a supportive mechanism [11].

In the endeavor to substitute the need for grafts and alternatively provide a supporting structure for the delivery of cells and biomolecules, nerve guidance conduits (NGCs) represent a viable strategy to bridge the gap of severed nerve injuries while guiding axonal and nerve regeneration [12]. Typically, the first- and second-generation NGCs are simply designed as tubular structures serving solely to bridge the nerve injury gap and physically direct the growth of axons [13]. Unfortunately, clinical investigations have shown that such nerve conduits are ineffective for those lesions greater than or equal to 3 cm [11]. In response, third-generation NGCs that better mimic the natural nerve microenvironment have been developed and are currently under extensive investigation. Increasing evidence has highlighted that close imitation of the morphological and physicochemical properties of the native extracellular matrix (ECM) of nervous tissues can achieve better therapeutic efficacy [14]. To maximize the axonal regeneration capacity, it is recommended that NGCs should recapitulate the essential attributes of native nerve environments through the biomimetic design [15] of the composition, structure, and function of nervous tissue [8]. Furthermore, NGCs should be able to deliver the desirable biochemical and physical cues via neurotrophic factors, ECM proteins, anisotropic gradients, and diverse types of supporting cells [13].

Hence, this review summarizes recent advances in the efforts toward the biomimetic design of NGCs. Firstly, the structural, compositional, and physical properties of PNS are reviewed to identify the key attributes that need to be considered in the design of a biomimetic NGC. Then, the review continues to elaborate upon the native nerve regeneration process with a particular emphasis on the role of NGCs in enhancing nerve repair. Furthermore, advances in materials, cells, and biofactors to improve the biomimicry and function of NGCs are discussed in great detail. In addition, suitable techniques used to fabricate biomimetic NGCs are evaluated in light of their ability to mimic natural nerves. Finally, this review concludes with future perspectives intended to inspire research that attempts to fully restore PNS functions and develop personalized NGCs. 

## 2. Physicochemical Properties and Regeneration Capacity of Peripheral Nerves

### 2.1. Structural, Compositional, and Physical Properties of Peripheral Nerves

Structurally, nerves within the PNS are organized in a hierarchal fashion to facilitate signal transduction from the CNS to peripheral tissues and organs (see Figure 1A,B). As the functional unit of the nervous system, neurons are specialized cells that transmit electrical and chemical signals to communicate with other cells at the synaptic junction [1]. They have three major components: (1) the cell body, where essential cell functions are performed, (2) the dendrites, which receive signals at the synapse, and (3) the axon, which acts as a cable to transmit electrical signals to the next synapse. The axon is typically covered by a myelin sheath, comprising lipids and proteins, to protect the axon and conduct electrical signals [16]. The myelinated axons, also called nerve fibers, vary in diameter from 0.2 to 20 µm, depending on their function and location [17], and are surrounded by the basal lamina. The basal lamina is a highly specialized ECM secreted by myelinating SCs, and it is primarily composed of type IV collagen and laminin crosslinked with proteoglycans such as heparin sulfonate [18]. The space between individual nerve fibers is called the endoneurium, and it comprises highly aligned type III collagen fibers in the same direction as the axons [19]. Nerve fibers are bundled into fascicles via a connective tissue layer called the perineurium. The size of fascicles differs between nerves and within the same nerve at different locations [20]. Fascicles are, further, bundled with a connective tissue layer called the epineurium to form a larger nerve. The epineurium acts as a protective layer for the nerve by shielding it from potential damage or injury and housing blood vessels (see Figure 1A) [21]. The thickness of the epineurium varies between 1 and 100 µm, depending on the nerve and its location [22].

In addition to neurons, the PNS contains several glial cells, such as SCs, satellite glial cells (SGCs), and neural stem cells, that maintain the extracellular environment and support the neurons. SCs protect the peripheral axons, regardless of myelination. During the myelination of the peripheral axon, an SC envelops a segment of the axon within its cytoplasmic groove and wraps around the axon to form the myelin sheath [19]. Unmyelinated axons are protected and supported by SCs that envelop and surround multiple unmyelinated axons to form small, shallow grooves in the nerve bundle. However, it requires a series of SCs to enclose an entire axon. SGCs are another functional cell type found in the PNS that envelop the cell bodies of neurons in the peripheral ganglia [19]. While their functions are not fully understood, SGCs are believed to play a critical role in neural homeostasis and have been studied in relation to pain [23]. Throughout the connective tissue of the nerves, neural stem cells can be found in the ganglia. These stem cells are typically dormant but can be activated after injury to aid in the regeneration of damaged tissue [17].

Another critical component of the PNS is the ECM, which supports and regulates the behaviors of residing cells and facilitates nerve regeneration. The ECM is primarily composed of type I, III, and V collagen fibers organized into a network that provides the structural framework of the nervous tissue. Distributed among the fibrous network are proteins such as laminin and fibronectin to support cell adhesion, growth factors to regulate cellular functions, and metalloproteinases to modulate ECM remodeling. In addition to these proteins, proteoglycans, and glycosaminoglycans (GAGs) can also be found in the ECM. Proteoglycans provide compressive strength and nutrient regulation, while GAGs such as hyaluronic acid (HA) hydrate and lubricate the ECM. Together, these components create an encouraging environment to support the development, maintenance, and regeneration of peripheral nerves [25].

Peripheral nerves experience various stresses during body movements such as stretching. As such, the PNS possesses specific mechanical properties crucial to its function and response to forces. Peripheral nerves are viscoelastic, enabling them to deform while dissipating energy. Typically, nerves can elongate 6% to 8% during normal body movement without damaging the tissue. However, straining the nerves by more than 11% but to an extent that is still within the toe region may trigger pain (See Figure 2). Upon exceeding 15%, the start of the elastic region, nerves begin to experience damage [17,18]. However, due to their viscoelastic nature, the maximal strain that nerves can withstand prior to injury depends on the strain rate [26]. Recent works modeling the viscoelastic behavior of axons demonstrated that changing the strain rate from 10% per second to 50% per second would change the axonal failure strain from 11% to 6% [27]. As such, an individual is more likely to experience significant nerve injuries during traumatic events such as car accidents or sports injuries than in the cases of stretching exercises at low strain rates. While nerves can tolerate tensile stresses, they are less tolerant of compressive stresses. As documented, compressive forces as low as 50 mmHg (6.67 kPa), even if only applied for 2 min, can cause axonal demyelination, while a higher compressive force of 300 mmHg (40 kPa) applied for 2 min can lead to degradation of the distal nerve fibers [18]. Additionally, the pathophysiological conditions of peripheral nerves closely regulate their mechanical behaviors. Injured nerves undergoing healing are typically stiffer and less elastic due to inflammation and fibrin deposition. However, after the nerve fiber is fully regenerated and the connective tissue surrounding it is remodeled, its compliance returns to normal.

### 2.2. Regeneration of Peripheral Nerve Injuries

Peripheral nerves can regenerate, but this capacity highly relies on the gap size of the injury. For gaps smaller than 1 cm, the nerve can heal with minimal assistance. For gaps larger than 1 cm, because of the complexity of coordinating different cells during the regeneration process, even a minor deviation can easily result in unsuccessful healing and loss of functions [1].

Following a PNI involving axonal disruption or nerve transaction, inflammation and degeneration occur at the proximal and distal ends to clear the damaged tissue. Nerve regeneration typically undergoes several stages and is highly orchestrated via cytokines and growth factors [20]. In the inflammatory and degenerative phase, tumor necrosis factor-alpha (TNF-α) and interleukin-1β (IL-1β) contribute to the recruitment of immune cells and start the inflammatory response. Interleukin-6 (IL-6) promotes the proliferation and activation of immune cells, facilitating debris clearance and initiating tissue repair [17]. Damage to axons leads to the disruption of axonal integrity, which causes the degeneration of nerve fibers located distally to the injury site through a process known as Wallerian degeneration (WD) (see Figure 3) [21]. Notably, SCs and macrophages infiltrate the damaged region to remove axonal fragments, myelin remnants, and tissue debris. This degenerative process often leads to the shrinkage of the distal nerve and the retraction of axon terminals from their corresponding tissues [22].

During the proliferation stage, the second step toward nerve regeneration, SCs proliferate and align to form Büngner bands to guide the axonal sprouting from the severed proximal end to their distal target. In this stage, key cytokines and growth factors such as transforming growth factor-β (TGF-β), fibroblast growth factor (FGF), and vascular endothelial growth factor (VEGF) are involved in stimulating the migration and proliferation of neural fibroblasts, SCs, and endothelial cells [9]. A temporary bridge of fibrin, i.e., a fibrous matrix produced by the crosslinking of fibrinogen [30], fills the gap between the proximal and distal nerve stumps and supports the migration of SCs, fibroblasts, and endothelial cells from both ends of the injured nerve. These cells orient themselves along the fibrin fibers, creating a biological tissue bridge within the gap. Without the formation of such a bridge, which likely occurs in nerve gaps greater than 1 cm, limited migration of SCs into the injury site would result in reduced formation of the glial bands of Büngner that provide trophic support and topographical guidance for axon regeneration [30].

At the last stage of nerve regeneration, i.e., the remodeling and maturation stage, axonal sprouts grow through the Büngner bands until they reach their target tissues. Once the connection is established, the regenerated axons undergo myelination, which is essential to functional recovery. In this stage, nerve growth factor (NGF), glial-derived neurotrophic factor (GDNF), and brain-derived neurotrophic factor (BDNF) support neuronal survival and axonal growth [9]. Typically, the axonal regrowth rate from the proximal end is roughly 1 mm/day [22]. In gaps larger than 1 cm, axonal sprouts cannot traverse the injury site, leading to disorganization and the subsequent loss of function. In this regard, NGCs can help bridge the gap and, consequently, enhance the chances of successful axonal regrowth across the injury site. Furthermore, NGCs can provide a defined path to prevent random axonal sprouting and guide neurite outgrowth toward functional recovery.

## 3. Design Considerations for Biomimetic Nerve Guidance Conduits

NGCs have undergone an evolution from simple, hollow-tube designs to multi-channeled structures and from nondegradable materials to bioactive, degradable materials. For example, first-generation conduits were solely meant to act as barriers against connective tissue infiltration while guiding directional nerve regrowth [31]. These conduits typically comprise decellularized allogenic or xenogeneic tissue or non-biodegradable materials such as silicone or polytetrafluorethylene (PTFE). However, decellularized grafts rely on scarce and costly sources, and non-biodegradable materials could cause complications, such as fibrotic encapsulation. Thus, the design of second-generation nerve conduits has focused on improving the biocompatibility of the implants via utilizing resorbable materials that are semi-permeable. Still, the simple, hollow tubular design of second-generation NGCs yields subpar functional regeneration compared to autologous nerve grafting. In recognition of the limitations of these designs and the growing body of evidence demonstrating the influence of extracellular features such as material stiffness, hydrophilicity, topography, and chemical properties on neural cell processes [32,33], emerging efforts are being made to formulate an encouraging microenvironment to improve functional recovery. As such, there is increased interest in designing NGCs that can recapitulate the key features of the native ECM of the PNS [6,30,31]. Diverse attempts have been made to fabricate NGCs that mimic various aspects, such as the composition, structure, and mechanics of nervous tissue.

### 3.1. Compositional Considerations

Inspired by the unique composition of native nervous tissue, incorporating materials, biomolecules, and cells found in nerves can be an effective strategy to introduce biomimetic design into NGCs [34,35].

#### 3.1.1. Material Choice

Nerve conduits can be made from various materials, including synthetic, natural, or hybrid biomaterials [36,37], and each type of material has its advantages and disadvantages (see Table 1). For example, nerve conduits made from collagen and poly (lactic-co-glycolic) acid (PLGA) have already received approval for the clinical treatment of peripheral nerve injuries because of their degradability and good biocompatibility. However, these grafts only partially meet clinical needs due to their limited effectiveness in promoting nerve repair, as well as the associated costs [36]. Compared to natural materials, synthetic polymers offer several benefits, such as ease of modification, minimal risk of pathogenic infection, low toxicity, and, presumably, fewer immune responses [38]. However, these materials often fail to provide biological guidance, as they lack the functional motifs found in the native ECM of nervous tissue. Conversely, natural materials are appropriate for cell growth and adhesion but may cause uninvited immune responses and are less reproducible [39]. While the materials used to create NGCs have been extensively reviewed [36], the materials with the most biomimetic features are highlighted below.

##### Biomimetic Synthetic Materials

Typically, synthetic materials are chosen for their ability to mimic the mechanical or conductive properties of native nerves [64]. To accommodate the mechanical complexity of native nerves, efforts have been made to combine various synthetic polymers, such as polylactic acid (PLA), polyglycolic acid (PGA), poly(lactic-co-glycolic acid) (PLGA), and polycaprolactone (PCL), to create multilayered NGCs with varying mechanical strength and degradation profiles [65]. For example, the combination of electrospun PLA, PGA, and PLGA nanofibers forms a layered NGC with a radial gradient degradation that displays similar or superior porosity, water absorption capacity, mechanical properties, and biocompatibility compared to single material NGCs (see Figure 4A) [66,67]. Such multilayer structures present noticeable advantages, allowing for better tissue ingrowth as the inner layers degrade while the slowly degrading outer layers effectively prevent NGC collapse and connective tissue invasion.

Considering the conductive nature of nerves for electrical signals, synthetic conductive polymers have received particular attention for their use in fabricating NGCs. Emerging evidence has shown that incorporating conductive polymers into NGCs promotes axonal outgrowth [68,69], aids in establishing connections with neuronal circuits, and enhances nerve impulse conduction to encourage nerve regeneration [70]. Among several representative conductive synthetic materials, such as polypyrrole (PPy), polyaniline (PANI), poly(3,4-ethylenedioxythiophene) polystyrene sulfonate (PEDOT:PSS), and reduced graphene oxide (rGO), PPy has been demonstrated to be a popular choice for bioelectric applications such as electrode-cell interfaces in bionics [41] (see Figure 4B). The characteristics of PPy, such as its biocompatibility, conductivity, dynamic nature, and ease of polymerization, make it an ideal candidate for creating NGCs. PANI is another material that is commonly employed to create conductive NGCs. Studies have shown that PANI is beneficial in promoting cell attachment, proliferation, and differentiation [44], and PANI-coated intraluminal microtubes within NGCs favor nerve regeneration [43]. Lately, PEDOT:PSS has also gained substantial recognition and growing interest in biomedical applications for its unique properties [71]. As a polyelectrolyte complex, PEDOT:PSS has excellent conductivity and can disperse in an aqueous solution to form hydrogels, allowing for easy processing and application [45]. In an effort to further demonstrate the importance of conductivity, NGCs coated with carbon nanotubes (CNTs) or PEDOT:PSS were used to regenerate the recurrent laryngeal nerve defects in a rabbit model, and it was found that the conductive NGCs could help improve vocal cord mobility and reduce thyroarytenoid muscle atrophy [46]. Recent work from our group has also demonstrated that rGO-coated poly(l-lactide-co-ε-caprolactone) (PLCL) NGCs can induce neuronal-like network formation under electrical stimulation [40]. All these findings indicate that recapitulating the conductivity of nerves in NGC design is essential for PNI regeneration.

While synthetic materials have demonstrated their advantages with tunable mechanical and degradation properties, and/or with conductivity, these materials normally exhibit subpar biological performance, especially compared to naturally derived materials. As such, these materials oftentimes need to be functionalized with biomolecules in order to achieve those desirable biological activities. Moreover, the degradation rate of many synthetic materials could not match well with native tissue ingrowth. Lastly, many of these materials have elastic moduli that are too high, making them unable to elongate 6–10% under a load below 2 MPa, which is typical for native nerves [18].

##### Biomimetic Natural Materials

To take advantage of the inherent bioactivity of natural polymers, many materials, such as chitosan, collagen, silk fibroin, gelatin, HA, and fibrin, have been used to fabricate NGCs. These materials can create an environment that facilitates and promotes cellular activities and supports cell-material interactions that are favorable for nerve regeneration [35].

Among various natural polymers, ECM-derived biomaterials, such as collagen, gelatin (hydrolyzed collagen), and fibrin, are commonly used in particular consideration of their presence in native nerve ECM. As the primary component of nervous ECM, collagen has been extensively explored to fabricate NGCs. A recent study showed that NGCs comprising collagen and chondroitin-6-sulfate and incorporated with fibronectin and laminin biomolecules enhanced the proliferation of SCs in vitro and regenerated the injured rat sciatic nerve in a similar capacity to autografts [49]. Apart from collagen, fibrin has also been used to fabricate nerve conduits with special regard to its self-assembly into oriented, cable-like networks for SCs to attach and migrate from the proximal to the distal end of an injury [30]. This has been further evidenced by a recent report on the use of an aligned fibrin nanofiber hydrogel, showing the rapid cell adhesion and migration of SCs in vitro, and in vivo regenerative outcomes close to those of autologous nerve grafts in the rat injury model [40].

Despite the noticeable advantages, ECM-derived materials often face challenges, such as high manufacturing costs and large source-dependent variabilities. In response, there is a continuous motivation to seek alternative ECM-like materials, such as silk fibroin (SF) and chitosan (CS). With a degradation rate close to the rate of nerve tissue ingrowth [63], SF-based NGCs displayed a lower risk of inflammation and rejection [72]. Moreover, testing of aligned SF NGCs in rat sciatic nerve models revealed both functional and morphological regeneration similar to those achieved with autografts [63]. Due to the presence of free amino groups, chitosan can interact strongly with some ECM proteins, such as laminin, fibronectin, and collagen, and help cell adhesion [36]. A recent study has shown that chitosan patterned with microchannels and nanofibers could promote SC migration and neurite outgrowth in vitro [73].

Moreover, more efforts have also been geared towards the combination of ECM-derived and ECM-like natural materials to mimic the composition of nervous tissue without incurring costs. One example is a chitosan–collagen hydrogel conduit integrated with SCs to promote peripheral nerve regeneration [74], in which chitosan stabilizes the structure via electrostatic interactions with collagen. While chitosan improved the mechanical strength of the conduit compared to collagen alone, collagen increased SC proliferation to support axonal outgrowth. In another study, gelatin and alginate were combined to develop anisotropic hydrogels with self-assembled capillaries for axonal ingrowth [52]. The addition of gelatin not only improved the biocompatibility but also increased the diameter of self-organized alginate microcapillaries (10–60 µm in diameter) by nearly 50%, leading to higher levels of SC migration and axonal ingrowth.

##### Biomimetic Hybrid Materials

Hybrid biomaterials, a subset of composites combining the biological properties of natural materials with the mechanical and functional properties of synthetic materials, offer a wide range of benefits, including tunable mechanical and degradation properties, scalable production, ease of processing, and biological activity [75]. There are many variations of hybrid materials, but recent interest in mimicking both the composition and function of nerves has led to the development of hybrid NGCs that contain two to three types of materials: (1) natural materials to improve cell adhesion and migration, (2) conductive synthetic materials to mimic the function of nerves, and (3) an optional synthetic material to provide mechanical strength.

In an effort to fabricate hybrid conductive conduits, PEDOT:PSS has often been included together with other materials, given that PEDOT:PSS can be made into various forms, is mechanically robust, and is highly elastic. For example, a recent publication demonstrated that a gelatin–PEDOT:PSS hydrogel was conductive and biocompatible, and was able to prolong astrocyte growth [76]. While this combination has not been used in NGCs, the encouraging results in supporting the cells of the CNS also imply its potential for PNI regeneration. Another hybrid NGC is the combination of SF with PEDOT:PSS to create a two-layer structure, of which the inner layer contained aligned electrospun SF and the outer layer comprised random electrospun SF, with an integrated network of self-assembled PEDOT:PSS throughout both layers for electrical conductivity. In vitro testing demonstrated that adding PEDOT:PSS to create a hybrid NGC did not affect biocompatibility but improved neurite outgrowth when compared to SF-only NGCs [77].

rGO is another popular choice for creating hybrid conductive NGCs due to its high conductivity, mechanical properties, and ability to be coated onto or dispersed into other materials. For example, NGCs fabricated via coating rGO on M. Menelaus butterfly wings effectively promoted the axonal regrowth and functional regeneration of nerve and muscle tissue compared to non-conductive controls [68]. Also, rGO-coated NGCs can combine electrical stimulation for in vivo application, showing enhanced functional regeneration compared to non-conductive NGCs and conductive NGCs without electrical stimulation in rat median nerve injury models [78]. Lastly, rGO can be dispersed into other polymers to create new conductive hybrid materials for the creation of conductive conduits (see Figure 4B,C) [2,79,80].

To better tailor the mechanical properties, additional materials can also be included in the mixture. For example, in a recent study on the creation of a hollow NGC, PLCL with good strength and toughness was added to SF to improve the flexibility and strength of the electrospun nanofibers of a conduit, which was coated with PPy [46] (see Figure 4E). This hybrid conductive NGC improved the early proliferation of SCs post-surgery and enhanced myelin formation, and most importantly, it was demonstrated to be effective in restoring sciatic function, similar to autologous graft, 12 weeks post-surgery. Another example was seen in the effort by Pillai et al. [81], who developed a multi-material-based NGC by using knit silk filaments coated with a mixture of PCL and carbon nanotubes (CNTs) to form the conductive interior tube and providing additional mechanical support to the NGC with an outer layer made of a hybrid of SF and polyvinyl alcohol (PVA). In vivo testing with 2-cm-gap rabbit models showed that these conduits could bridge the nerve gap within 30 days post-implantation without further biomolecule functionalization.

**Figure 4 nanomaterials-13-02528-f004:**
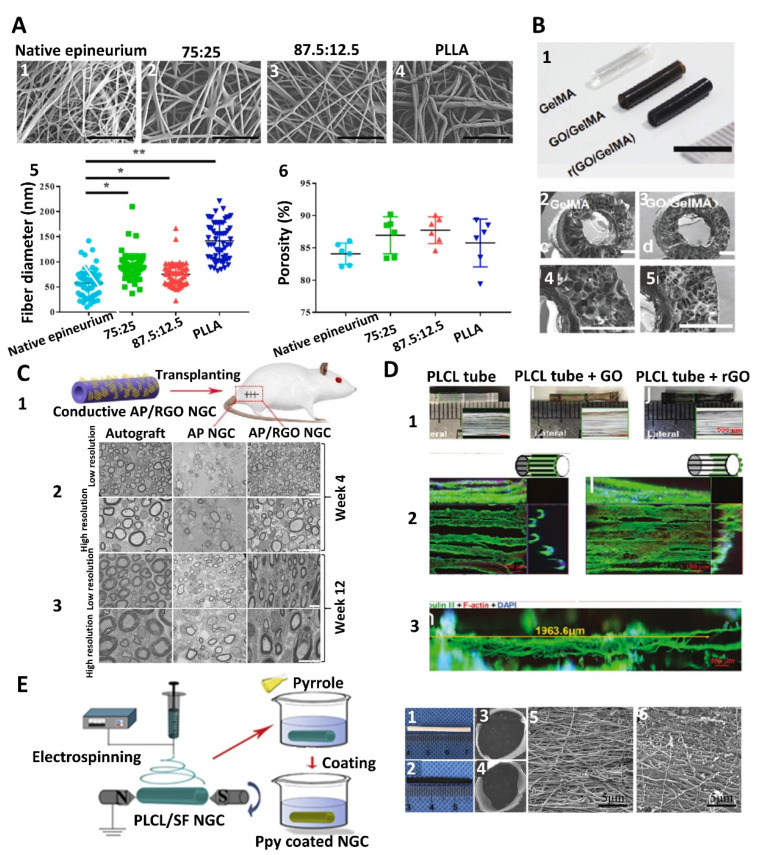
Hybrid materials are used to create biomimetic NGCs. (**A**) (1–4) SEM images of native epineuria and electrospun nanofibers made from PLLA and gelatin combinations at different ratios; (5,6) characterization of fiber diameter and porosity, showing that a hybrid of PLLA and gelatin could better mimic the properties of the native epineurium [67]. (**B**) (1) Fabrication of conductive GelMA-based NGCs via the inclusion of reduced graphene oxide (rGO); (2–5) SEM images of the cross-sections of GelMA and GO/GelMA conduits, indicating the presence of permeable pores in the conduit wall [2]. (**C**) In vivo performance of hybrid SF/rGO NGC with improved regeneration capacity of the injured sciatic nerve [79]. (**D**) (1) Schematic illustration of the development of a 3D tube composed of longitudinally aligned PLCL microfibers coated with rGO; (2,3) confocal microscopic images of outgrown neurites from PC12 cells along the rGO-coated microfibers [40]. (**E**) Schematic illustration of a conductive NGC made of electrospun PLCL and SF hybrid and then coated with PPy. (1,2) Light microscopic images and (3–6) SEM images of PPy-coated PLCL/SF NGC [46].

#### 3.1.2. Cellular and Biomolecular Choices

Despite the advances of hybrid materials in promoting nerve regeneration, NGCs can further benefit from the incorporation of cells and biomolecules, which would deliver more stimulating cues similar to native nerves for better axonal regeneration [49]. Thus, a growing effort has been made to encapsulate PNS-associated or non-associated cells, growth factors, and ECM proteins into NGCs to promote cell adhesion, SC migration, vascular infiltration, axonal growth, and, consequently, functional recovery.

##### Cellular Choices

In recognition of the roles of SCs and neural stem cells in supporting, promoting, and guiding axonal sprouting following PNIs, patient-derived cells have been incorporated into NGCs to minimize the risk of immune rejection [82], promote axonal regrowth, and provide neurotrophic support [73]. For example, Liu et al. demonstrated that collagen-based conduits loaded with neural stem cells returned functional levels in the regeneration of 3-mm rat spinal cord injury models [83]. This is very promising, given that spinal cord injuries are more difficult to regenerate than PNIs. However, neural stem cell transplants alone cannot regenerate PNIs, as evidenced by the outperformance of neural stem-cell-seeded NGCs over the cell transplant-only group in a mouse sciatic injury model [84]. Furthermore, the role of SCs in nerve regeneration should not be understated, as SCs encapsulated in NGCs could secrete up to 6 pg/mL of nerve growth factor (NGF), stimulating axonal sprouting in neurons without further exogenous support [74]. In this regard, combining both cell types, i.e., SCs and neural stem cells, in NGCs might be essential to improve functional recovery for PNIs, as evidenced by a recent study showing that the co-transplantation of SCs and neural stem cells with laminin–chitosan–PLGA NGCs significantly increased brain-derived nerve factor (BDNF) levels in rat models and resulted in comparable functional recovery in 7-mm recurrent laryngeal nerve segments compared to nerve autografts [85].

Despite the encouraging results from the encapsulation of SCs and neural stem cells in NGCs, it remains challenging to harvest and expand such autologous cells for clinical use. As such, Schwann-like cells derived from mesenchymal stem cells (MSCs) have been considered a promising alternative [11,86]. Results demonstrate that MSCs enhance the expression of neurotrophic and angiogenic factors that can better emulate the native environment for nerve regeneration [86]. For example, including adipose-derived MSCs into laminin-coated chitosan/PLGA NGC containing gold nanoparticles and BDNF helped return sciatic function to 80% of the uninjured control and achieved the least amount of muscle atrophy compared to the groups without MSCs [87]. Additionally, the ability of MSCs to express VEGF is believed to play a vital role in nerve recovery, as demonstrated by the improved myelination level and increased axonal diameter in rat facial nerve models treated with NGCs containing dental-pulp-derived MSCs that overexpressed VEGF [88].

In addition to MSCs, SCs derived from induced pluripotent stem cells (iPSCs), which are generated from autologous somatic cells, represent another promising cell source to be included in NGCs. Unfortunately, the current method of deriving SCs from iPSCs cannot generate enough viable cells [89], thereby relegating this source for use as a model to study the functions of human SCs [90].

##### Biomolecular Choices

Given that the primary purpose of encapsulating cells in NGCs is their ability to secrete growth factors and ECM proteins, oftentimes, reaching the desired therapeutic concentration is challenging. To this end, incorporating biomolecules directly into NGCs would allow for more control of the concentration needed, along with reductions in cost, time, and the risk of immune rejection associated with cellularized NGCs.

The typical biomolecules involved in nerve regeneration are neurotrophic factors (NTFs), such as NGF, BDNF, or glial-derived neurotrophic factor (GDNF), which promote neural cell survival and guide axonal growth [91]. Since NTFs are present at different stages and varying concentrations during nerve regeneration, the regeneration capacity and rate are closely regulated by the initial concentration, dosage, and release kinetics of NTFs [92]. For example, an NGC with gradient NGF concentrations from early burst release and with gradually released BDGF from gelatin nanoparticles has led to functional recovery and myelination similar to autografts, as NGF-induced SC infiltration and BDGF promoted myelination in the later stages [93].

Recently, exosomes secreted by SCs have also been incorporated in NGCs or used to guide MSC differentiation for NGC use. These SC-secreted exosomes contain proteins, lipids, and nucleotides necessary for stem cell differentiation into Schwann-like cells that can be used in NGCs [94]. Furthermore, works by Hu et al. and Namini et al. demonstrated that the incorporation of exosomes within NGCs as biomolecules promotes PNI regeneration in rat models [95,96].

ECM proteins such as fibronectin and laminin are commonly coated or integrated within NGCs since they could improve SC attachment and migration while promoting neurite outgrowth [97,98]. A recent study demonstrated that incorporating fibronectin, laminin-1, and laminin-2 at a ratio of 1:4:1 into NGCs could achieve high levels of NGF, GDNF, and VEGF secretion by SCs while minimizing the secretion of pro-inflammatory cytokines [49]. Further testing in a 15-mm sciatic gap injury rat model revealed that an NGC with a 1:4:1 ratio of fibronectin, laminin-1, and laminin-2 exhibited a significant increase in the number of axons within the conduit and higher levels of vascular infiltration compared to autologous nerve grafts 8 weeks post-implantation.

It is unmistakable that the composition of NGCs has a profound effect on their overall nerve regeneration efficacy. The material of the conduit serves as the foundation to offer those physicochemical properties desirable for cell attachment, mechanical strength, and conductivity. The incorporation of cells and biomolecules would further increase the biomimicry of NGCs by providing stimulating factors to support axonal outgrowth. Such an enhancement could be culminated via the inclusion of those ECM proteins to facilitate cell adhesion and migration into the NGCs and, subsequently, the stimulation of the secretion of NTFs for nerve regeneration.

### 3.2. Structural Considerations

Apart from the essential regulation endowed via NGC composition to recapitulate the native environment, the fabrication of NGCs with various configurations would also affect the nerve-regenerating performance of conduits.

#### 3.2.1. Biomimetic Architectures

To better guide axonal growth and support nerve regeneration, it would be desirable for NGCs to capture the microstructural features of native nerves [99]. In recognition of the drawbacks of non-porous, hollow NGCs, such as the limited capacity of bridging nerve gaps over 1 cm and poor permeability to nutrients and growth factors [100], secondary structures such as grooves, pores, channels, and fillers have been added to NGCs [101]. The creation of aligned grooves or channels in NGCs could effectively mimic the fibrin network and Büngner bands for axonal regeneration, while the presence of fillers and pores supports cell infiltration, neovascularization, and nutrient diffusion to the regenerating nerves [102] (see Figure 5).

##### Biomimetic Topographical Features of NGCs

The inclusion of additional features in the luminal wall of NGCs, such as pores to increase permeability and microgrooves to provide alignment cues, would significantly improve the efficiency of hollow NGCs in nerve regeneration. For example, a 3,4-dihydroxy-l-phenylalanine (DOPA)-coated PLGA NGC with the presence of porous microgrooves on the luminal wall enhanced the permeability of nutrient exchange and promoted axonal outgrowth [113]. Notably, such aligned porous microgrooves could modulate neurite orientation and the length of neural mouse stem cells in vitro compared to porous flat NGCs. Furthermore, compared to the non-patterned NGCs filled with random or aligned fibers, the porous micropatterned NGCs dramatically rebridged the neurofilaments and increased the sciatic function index, onset-to-peak amplitude, muscle weight, and the fiber diameter 8 weeks after implantation in a rat sciatic model. This likely occurred due to the presence of grooves in NGCs that mimic the fibrin network and Büngner bands, increase the surface area, and improve cell attachment [114].

The pores within the luminal walls of NGCs can also be used to load biomolecules for local release to induce the preferred cellular response. A hollow NGC comprising 3D-printed gelatin-hydrogel-based microgrooves wrapped with electrospun PLCL membrane successfully guided axonal regeneration while releasing a neurogenic drug from the gelatin hydrogel [14]. Testing in rat sciatic models showed that such conduits were able to guide axon outgrowth and achieve remyelination, which was essential for functional recovery. Apparently, NGCs with aligned pores play a multifaceted role in maintaining the sustained release of a neurogenic drug and guiding axons to their target destinations [115]. It is also noted that porous, hollow NGCs with thicker walls (e.g., 500 μm) exhibited a better improvement in nerve regeneration capacity since they could achieve a suitable trade-off between porosity and stiffness to ensure the formation of a fibrin cable in the early regeneration stage while maintaining adequate mechanical properties, nutrient permeability, and slow biodegradation to protect the nerve. In rat sciatic injury models, thicker porous, hollow NGCs resulted in a higher density of Schwann cells and myelinated axons than thinner walled tubes [116].

##### Biomimetic Fillers for NGCs

Hollow NGCs, regardless of their added topographical features, still fail to capture the key structural attributes of native nerves, that is, containing multiple fascicles that each house numerous axons. In this regard, a growing interest has shifted to filling the luminal space of NGCs with various fillers, especially the aligned anisotropic ones, closely mimicking the conditions of the endoneurium surrounding individual axons to improve the regenerative capacity of nerves.

There is growing interest in utilizing natural polymers as NGC fillers in the forms of hydrogels, filaments, or porous sponges. Due to their soft properties and biocompatibility, these materials can serve effectively as regenerative guides for repairing PNIs and supporting the functionality of nerve conduits during the repair process [117]. For example, an NGC filled with the hydrogel of a recombinant elastin-like protein facilitated the formation of a tissue bridge between the proximal and distal nerve stumps of a 10-mm nerve gap with myelinated axons and innervations of distal muscle compared to an unfilled control [64]. Injectable hydrogel fillers can also be used in combination with hollow tubes to improve the regenerative capacity of NGCs. Filling a hollow NGC with injectable NGF-containing CS-HA prior to implantation effectively promoted nerve regeneration compared to hollow NGC controls in a rat sciatic model [118].

Compared to hydrogels, anisotropic aligned fillers are believed to better induce axonal organization than random fillers by encouraging axons to extend through the conduit while facilitating nutrient exchange. For instance, CS fibers or chitin-containing composite fibers are employed as fillers. These fibers guide axonal growth through their oriented structure to establish electrical connectivity between injured nerve terminals [119]. Similarly, a silk-based NGC with tunable anisotropic architectures and micropores favored nerve regeneration via supporting the proliferation of Schwann and PC12 cells in vitro and enabled the repair and functional recovery of sciatic nerve defects in rat models [91]. However, balancing porosity and structural alignment is challenging, as high porosity may lead to neuroma formation, while high levels of alignment may hinder the functional reconnection of the injured nerve [101]. Still, NGCs with a pore size of approximately 10–40 µm and porosity of roughly 80% are the most efficient at axonal regeneration in PNIs [120].

Fillers can also be functionalized with biomolecules or conductive polymers to achieve high biomimicry. The introduction of pores can aid in nerve regeneration and nutrient exchange; however, pores larger than 30 µm may lead to the infiltration of fibrous tissue, thereby blocking axonal regeneration. To this end, coating the fillers with fibrin and HA can help alleviate this issue [104]. Furthermore, adding peptides that mimic the activity of BDNF and VEGF to an aligned, hydrogel-filled NGC can noticeably improve nerve regeneration, myelination, conduction, and motor function recovery when bridging rats’ 15-mm sciatic gaps in comparison to hollow or filled NGCs [121]. Conductive hydrogel fillings also improved neurite outgrowth, as demonstrated via the anisotropic rGO-containing NGC, leading to a 47% longer neurite length compared to those without rGO [122].

##### Multi-Channeled NGCs

While filled NGCs are able to facilitate neuronal cell adhesion, promote cell differentiation, and guide neurite outgrowth [123,124,125], fillers, occupying the luminal space, would limit the number and size of axons growing through the conduit [126]. As such, multi-channeled NGCs have been proposed to enhance nerve regeneration within large nerve defects, as they can provide better guidance for axonal ingrowth than the pores of anisotropic fillers [93,126]. These channels also partially mimic the unique morphology and structure of peripheral nerve fascicles, where myelinated and unmyelinated axons are surrounded by a collagen-based endoneurium, to promote neurogenesis and nerve regeneration [62].

The primary difficulty in mimicking fascicles is the large variation in fascicle size and axon number, which are closely related to the location and function of the peripheral nerve [127]. Typically, the diameter of axons ranges between 4 and 16 μm [128], and fascicles can contain a few to several hundred axons [129]; thus, it remains highly challenging to identify an ideal channel size that effectively mimics the fascicle anatomy. Through experimentation, researchers have noticed that the channel diameter of ~125 μm seemed more favorable to encouraging functional regeneration [85]. Other comparative studies, to a certain degree, have corroborated this evidence. For instance, in vitro studies have shown that smaller-diameter channels (~180 µm) enhanced astrocyte alignment [130], and NGCs with channel diameters between 200–300 µm induced the ingrowth of aligned axons, glial cells, and vasculature in a 10-mm rat sciatic model [131]. Interestingly, conduits with large channel diameters (e.g., >450 µm) led to poor nerve regeneration in a rat spinal cord, showing reduced numbers of regenerative axons two months post-implantation [132], and the NGCs with small-diameter, self-assembled microcapillaries (50–60 µm) exhibited the worst SC migration and axonal ingrowth compared to the larger-diameter microcapillaries (~80 um) [52].

Apart from the channel diameter, the number of channels in an NGC also impacts nerve regeneration. As expected, a higher density of channels could effectively reduce the dispersion of axons, therefore promoting more directed and organized axonal growth [2]. Similar to the situation with channel diameter, it is also difficult to determine what ideal number of channels is needed to better mimic the native nerve for PNI regeneration. Moreover, increasing the density of channels in NGC with a fixed size remains a challenge despite the technical advances in fabrication such as 3D printing and microfluidics, which offer more precise control over the development of small channels at a high density [13,133,134]. In general, it would be more practical to consider the location of the injury and then develop the biomimetic, multichannel NGCs based on the fascicle size and axon density of the local nerve (see Figure 6A–C).

#### 3.2.2. Fabrication of Biomimetic Features

Many fabrication technologies, such as electrospinning, 3D printing, molding, casting, freeze-drying, and micropatterning, are available to deliver some of the biomimetic features to the above-discussed NGCs (Figure 4B) [2,135]. Depending on the fabrication method, there is a good possibility of controlling the porosity, channel size, density, and alignment, but each method has its limitations (see Table 2, Figure 7).

As shown in Figure 7, the topographical features of NGCs can be implemented via electrospinning, inkjet printing, and micropatterning. Electrospinning, a high-electric-field-driven spinning method, has received great attention and has become a popular platform for fabricating fibers with diameters in the nano-to-micrometer range, similar to the size of collagen fibrils found in the native ECM [46]. In this process, the high electric field applied to a polymer solution or polymer melt generates a jet that pulls the polymer into thin fibers [15,41,137]. By varying the polymer concentration, electric field, and collection substrate, the fiber diameter, fiber organization (aligned vs. random), and the thickness of fiber matrices can be modulated [138]. Aligned fiber matrices can then be rolled to create NGCs that contain aligned topographical cues for guided axonal outgrowth [40,111]. For instance, an NGC containing layers of electrospun nanofibers, GO/PCL microfibers, and PCL microfibers was able to provide topographical cues, permeability, mechanical stability, and electrical conductivity to guide Schwann cells and neuronal growth [139].

Inkjet printing and micropatterning are less utilized to create NGCs than electrospinning, but both techniques enable control over the local topography. Inkjet printing, a subtype of 3D printing, uses heat or piezoelectric pulses to jet bio-inks in a drop-by-drop fashion. This method can create spatial and concentration gradients of various biomolecules, print conductive patterns, and localize cells onto NGC surfaces [15]. Due to the ready incorporation of growth factors, cells, or conductive nanoparticles into inks for inkjet printing, it is very likely to achieve the bioactive topographical modifications of NGCs and specifically promote neurite outgrowth [14,140,141]. On the other hand, micropatterning is a template-based method that uses photolithography, microcontact, or microfluidic stamping to create a variety of micropatterns, including lines, dots, and grids, to guide axonal alignment and promote neurite outgrowth [142,143,144]. However, similar to electrospinning, inkjet printing and micropatterning are also done on 2D surfaces, which generally require the rolling step to create conduit structures.

Given that a pore size between 10 and 40 µm and a porosity of roughly 80% are typically needed to recapitulate the native nerve environment and encourage PNI regeneration [120], efforts have been geared toward the creation of NGCs using freeze-drying or solvent casting/salt leaching. Freeze-drying is often employed to create porous conduits by sublimating the solidified solvent (e.g., ice) between polymer molecules and leaving the interstitial space empty [104]. The porosity of the NGC can be easily customized by adjusting the polymer concentration, and the pore morphology can be tailored by varying the freezing speed and methods. Although its ability to create interconnected anisotropic pores within NGCs has been recognized [49], the freeze-drying method has limited control over the size and uniformity of pores. In this regard, efforts have also been shifted to take advantage of solvent casting and salt-leaching as an alternative method, allowing for better tailoring of the porosity and pore size of NGCs. In this method, the polymer solution is mixed with porogens, typically salts such as sodium chloride or calcium carbonate, and cast into a mold. Upon solvent evaporation, the salts are leached out, leaving pores behind [145]. The pore size and distribution are highly dependent on the salt templates, determining the poor control of pore distribution; meanwhile, the pores might not be well connected, which is unfavorable for nerve regeneration. Generally, this method can create pores from 15 to 500 μm and porosities between 30% and 90% [146,147].

Three-dimensional printing and molding have been exploited to create multichannel NGCs. Three-dimensional printing, particularly stereolithography (SLA) and fused deposition modeling (FDM), offers a platform to fabricate complex structures with the opportunity to encapsulate living cells and biomolecules [94,96]. For example, the SLA-enabled fabrication of NGCs from gelatin–methacrylate (GelMA) demonstrated the possibility of creating multi-channeled NGCs with varying diameters and densities [57]. However, constrained by the printing resolution and time consumption, molding becomes a primary choice for fabricating multichanneled NGCs. Generally speaking, conduits made via molding tend to have a uniform channel shape and size and are easily scaled up for mass production. Furthermore, if molding is employed without additional pore-generating techniques but using compatible biomaterials, such as GelMA, decellularized ECM, collagen, or alginate, it is possible to create multichanneled, cell-encapsulated hydrogel NGCs [148]. However, molding in combination with pore-generating techniques might face the challenge of not being able to encapsulate cells, but this can be resolved via seeding with cells post-fabrication to reinforce the nerve regeneration capacity [74,149].

**Table 2 nanomaterials-13-02528-t002:** Advantages and limitations of fabrication methods used to create biomimetic NGCs.

**Fabricating Topographical Features**			
**Technologies**	**Advantages**	**Disadvantages**	**Ref.**
Electro- spinning	Control alignment to mimic PNS ECM Creates high-density, small pores, similar to native neural ECM Enables the incorporation of key growth factors and biomolecules	Requires materials that can dissolve in volatile solvents Low controllability of fiber diameter Generates 2D matrices to be rolled into NGCs	[13,14,49,105,123,135]
Inkjet printing	High control of biomolecule or cell localization Enables spatial and concentration gradients of biomolecules Customizable designs High resolution (~50 µm)	Limited printable materials Biomolecules are only coated onto surfaces May affect the bioactivity of biomolecules and cell viability The 2D matrix needs to be rolled into NGC	[138]
Micropatterning	Precise and reproducible method Micron-scale patterns possible	Complex process required to develop micropattern templates Templates not modifiable after fabrication Biomolecules are only coated on surfaces The 2D matrix needs to be rolled into NGC	[102,125,142,144]
**Fabricating Porous Features**			
**Technologies**	**Advantages**	**Disadvantages**	**Ref.**
Freeze-drying	Can create random and isotropic/anisotropic pores Preserves bioactivity of natural materials and added biomolecules Technique applicable to many common biomaterials	Potentially reduces material’s mechanical strength Poor control over pore size and size distribution Molds required	[49,62]
Solvent casting and salt-leaching	Simple fabrication process Controllable pore size Maintains mechanical properties of the material used	Potential toxicity from solvent Requires materials that can dissolve in a volatile solvent Poor control over pore size and distribution Mold required	[15,150,151]
**Fabricating Multichannel**			
**Technologies**	**Advantages**	**Disadvantages**	**Ref.**
Molding	Precise and reproducible method The process is compatible with both natural and synthetic materials Easily scalable process Can create complex, multichannel structures	Limited structural variability Every design requires a separate mold	[15,56,120]
3D printing	Highly customizable process Complex geometries/multichannel designs Rapid prototyping Enables cell encapsulation together with multichannel fabrication Stereolithography has a high resolution	Limited material selection Limited resolution, depending on 3D printing method (fused deposition printing, hydrogel/cell printing)	[9,152]

To overcome the limitations of individual fabrication methods, it would preferably combine various methods or use hybrid technologies to fabricate biomimetic NGCs. For example, Liu et al. used near-field electrowriting, dip-coating, and electrospinning to create a tri-layer conduit with structural characteristics resembling the human nerve [56]. Recently, electrospinning, freeze-drying, and molding have also been combined to create aerogels that mimic the native nerve ECM, in which cut fragments of electrospun nanofibers are dispersed in a solution, crosslinked to form the slurry for molding and then lyophilized to create a 3D matrix that mimics the organization of the ECM [153]. Taking advantage of the ability to create anisotropic and interconnected pores through freeze-drying, the aerogels can be formed to have an aligned or hierarchical structure [153]. Although the utilization of biomimetic aerogels for NGCs has yet to be demonstrated, we did adopt a similar strategy to create channeled NGCs [62]. The development of radiative heating textiles has demonstrated the feasibility of creating SF–rGO-based aerogels with a hierarchal organization for the development of biomimetic conduits [153].

#### 3.2.3. Mechanical Properties of Biomimetic NGCs

In addition to composition and design, the mechanical properties of the NGCs should, preferably, match the stiffness, flexibility, and elongation of native nerves. Evidently, the NGCs that closely resemble the morphological, physical, and mechanical properties of native nervous ECM tend to achieve more effective therapeutic outcomes [143]. Since the elastic modulus of human peripheral nerves is approximately 0.5–13 MPa [154], NGCs made from soft polymers with an elastic modulus in this range or slightly higher have immense potential to heal PNIs. During normal body movement, the nerves can be stretched as much as 6 to 8%; thus, NGCs should be able to tolerate such a stretch under repetitive stresses [18].

## 4. Discussion and Future Outlooks

Since the first attempt to bridge the peripheral nerve gap using nerve tubulation in 1880, NGCs have evolved from simple, rigid, tubular structures to flexible, multichannel configurations with additional topographical features and cellular/biomolecular stimuli, which has greatly advanced PNI regeneration so that it is close to that of autografts. However, it remains far from optimal functional recovery. Even autografting, still considered the gold standard in nerve reconstruction, does not lead to full functional restoration, as its material is typically harvested from the sural nerve, a sensory nerve below the skin surface of the calf, to replace a motor nerve [154]. However, this observation does not align well with animal studies, in which the sciatic nerve with both sensory and motor nerves is transected and then sutured back to act as the autograft control [155]. As such, there is a disparity in autograft outcomes in clinical applications compared to animal studies. The autografts in animal studies fully mimic the transected nerve in composition, structure, and function, allowing for full functional recovery after surgery, whereas the mismatched size and function of autografts with the transected nerves in humans only lead to suboptimal recovery. In this regard, it becomes essential for NGCs to maximally recapitulate the key attributes of the injured nerves to better support regeneration. Unfortunately, NGCs currently face the same hurdle that makes autografts ineffective in functional recovery—not fully mimicking the native nerve tissue.

In response, various efforts have been made to possibly develop NGCs that can adequately mimic the composition, structure, and biological functions of native nerves. Hybrid materials, combining the inherent bioactivity of natural polymers with the functional mimicry provided via conductive synthetic polymers, represent a better option for fabricating biomimetic NGCs while maintaining the elastic modulus of 0.5 to 13 MPa [154]. Furthermore, the inclusion of patient-derived SCs or neural stem cells in biomimetic NGCs [156] not only delivers more biostimuli to guide and initiate nerve regeneration but also helps better integrate with nerve stumps for better functional recovery. Adding some vital biomolecules to the NGCs, such as fibronectin and laminin, can formulate a favorable microenvironment for SC adhesion, growth, and migration, which plays a key role in stabilizing and protecting newly outgrown axons [49].

Ideally, biomimetic NGCs should recapture the morphological uniqueness of the native nervous tissue matrix. In this case, it would be preferred for NGCs to possess small (10–40 µm in diameter) and anisotropic pores, accounting for 80% of the NGC volume. NGCs should contain microchannels similar to the size and number of fascicles found in the transected nerve. The luminal surface of these channels should be smooth and hydrophilic to encourage neurite spreading and outgrowth while containing aligned features to guide axonal outgrowth.

Another important aspect in designing biomimetic NGCs is the need to consider the conductive nature of nerves, especially recognizing the encouraging effects of electrical stimulation and conductive nerve conduits on the expedited growth of axons [157,158]. The application of electrical stimulation through a conductive conduit has the potential to activate neurite extension and initiate signaling cascades of neurotrophic factors. These processes can promote the regrowth and elongation of axons, as well as facilitate the polarization and remyelination of SCs [159]. Studies have also demonstrated that electrical stimulation regulates cellular activities, including cell adhesion, proliferation, migration, and protein production [160,161]. However, encoding the mechanistic regulation of neuronal cells via conductive NGCs remains elusive. Moreover, the mechanism through which electrical stimulation triggers nerve regeneration has yet to be fully understood, and different opinions have also been shared regarding the ideal electrical stimulation parameters for functional recovery.

Several technological challenges should be overcome, along with the continuous improvement of biomimetic NGC design. Firstly, many current conductive materials have limited biodegradability, in contrast to the fast loss of mechanical strength due to the fast degradation of many natural materials [43]. Thus, the development of biodegradable conductive materials would greatly meet the need for conductive NGCs. Secondly, in response to the foreseeable challenge in harvesting, culturing, and expanding patient-derived SCs, deriving SCs from iPSCs is a highly viable strategy despite its low yield [89]. Further improvements in differentiation efficiency would be much more beneficial for the establishment of cellularized NGCs.

In the development of intricate biomimetic NGCs, limited fabrication methods are considered the bottleneck. Despite promising advances in the formulation of biomimetic aerogels, molding is not effective in achieving structural complexity, as the extraction of the formed structure from the mold remains a challenge without structural distortion or destruction. 3D printing, which does allow the creation of intricate structures, has a poor printing resolution [155,162], which determines its difficulty in fabricating small channels (less than 100 μm) and high channel density within conduits. Moreover, it remains a major challenge to add aligned topographical features to the channels of small biomimetic conduits, especially in the case where many topographical fabrication methods require a 2D surface.

Throughout the process of reviewing the literature, we started to realize that most of the current research has not carefully considered the regulatory functions of vascularization in supporting the outgrowth of axons and, subsequently, nerve regeneration. As a matter of fact, blood vessels are a crucial component of nerve anatomy; so far, only very limited attempts have been made to design or study the role of vascularization in NGCs. VEGF, a growth factor responsible for angiogenesis, can be found during nerve regeneration, and several studies have highlighted the importance of SC-secreted VEGF in promoting nerve regeneration [49,88,163]. However, more evidence is necessary to show the benefits of directly incorporating VEGF or endothelial cells in NGCs to improve PNI recovery. Particularly, it would be highly intriguing to determine whether vascularization alone can substantially impact functional recovery. If so, the factors regulating vascular infiltration and angiogenesis should be considered during the design and fabrication of NGCs.

Apart from the endeavor of perfecting the biomimicry of NGCs, personalizing NGC design to specifically fit individual patients and injury sites would be beneficial for NGCs’ integration with hosts and better support of functional recovery. Given that the size of the nerve, the number of fascicles, and the number of axons within each fascicle are closely related to the anatomical location and nerve function [9,99], the current practice, i.e., designing a one-size-fits-all biomimetic NGC, would see many possible pitfalls. To address such a challenge, it would be necessary to establish a platform that enables the prediction of the size, structure, and mechanical features of any given nerve for each patient. This can likely be accomplished through developing algorithms that are trained via machine learning to predict the ideal biomimetic NGC configurations, such as mechanical strength, channel size, channel number, pore size, and porosity, for individuals. As proposed, such a model can be trained with images of various nerves in different locations of the human body and from different people to ensure prediction accuracy. Such an advancement would help increase the success rate of full functional recovery with biomimetic NGCs.

## 5. Conclusions

While peripheral nerves can regenerate, large gaps (>1 cm) typically result in unsuccessful healing and a loss of function. In view of the shortage of, and comorbidities associated with, autografting, NGCs serve as a promising alternative to bridge the gap of the severely damaged nerve and guide axonal growth.

To maximize the regenerative capacity of NGCs and restore lost nerve function following severe PNIs, it is preferred for these conduits to deliver a native PNS-like environment via the optimal selection of composition, structural configurations, and functionality close to the native nerves. With respect to composition, hybrid biomaterials combining both the biological attributes of natural materials and the mechanical/conductive properties of synthetic materials are the most attractive ones to achieve the desirable physical performance (e.g., flexibility, degradation, and conductivity) while inducing the required cellular responses (e.g., the recruitment of neuronal cells, the migration of SCs, and improved myelination). NGCs can further benefit from the incorporation of cells such as SC or neural stem cells and biomolecules, such as fibronectin, laminin, NGFs, and/or SC-secreted exosomes, to better mimic the composition of native nerves and promote cell adhesion, SC migration, vascular infiltration, axonal growth, and consequent functional recovery. To mimic the structural hierarchy of PNS, where each nerve houses the fascicles and each fascicle holds the nerve fibers to facilitate signal transduction from the CNS to peripheral tissues, biomimetic NGCs are also designed to include additional features, such as grooves, pores, channels, and fillers. Parallel grooves and channels should present nano- and micro-scale topographical cues similar to the fibrin network. Additionally, NGCs should contain fillers and pores (10–40 µm in size with 80% porosity) to support cell infiltration, neovascularization, and nutrient diffusion to facilitate nerve regeneration. Finally, conduits should have multiple channels whose size and density maximally mimic the fascicles at the injury site.

To implement the structural complexity of biomimetic NGCs, various fabrication technologies, such as electrospinning, 3D printing, molding, casting, freeze-drying, and micropatterning, can be adopted as either single fabrication platforms or combined ones to create those biomimetic features. Advances in high-resolution fabrication, such as e-beam lithography, would further enhance the capability of creating NGCs with delicate structures.

Future research would benefit from the fabrication of NGCs that more closely mimic the native ECM, providing the necessary cues for neural regeneration and simultaneously promoting functional recovery. Such endeavors can further propel the development of personalized NGCs with the potential to fully integrate with host tissues. This would also benefit from the consideration of the roles of vascularization and electrical stimulation in facilitating nerve regeneration. Ultimately, the development of NGCs that can support the functional recovery of PNIs has the potential to revolutionize clinical practice and improve the quality of life for millions of patients worldwide.

## Figures and Tables

**Figure 1 nanomaterials-13-02528-f001:**
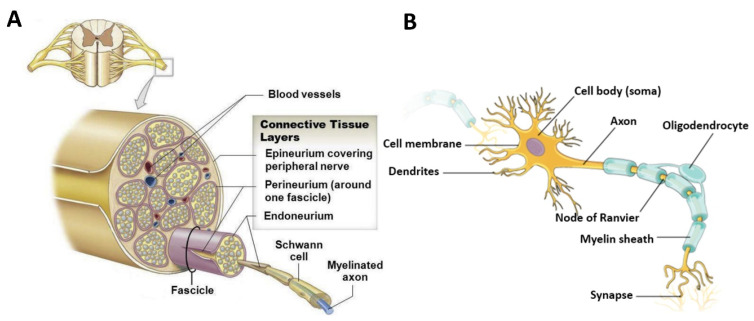
(**A**) A cross-sectional view of the nerve shows the hierarchy of the peripheral nerve [23]. (**B**) The anatomical structure of a neuron, reproduced from [24].

**Figure 2 nanomaterials-13-02528-f002:**
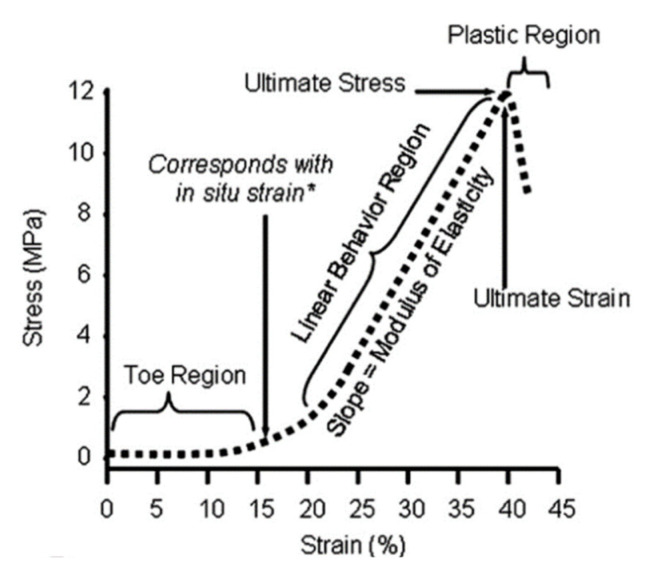
Stress–strain curve of a typical peripheral nerve, reproduced from [18]. This graph was generated based on the data collected in situ from rabbit tibial nerves using a strain rate of 0.5% per second [28].

**Figure 3 nanomaterials-13-02528-f003:**
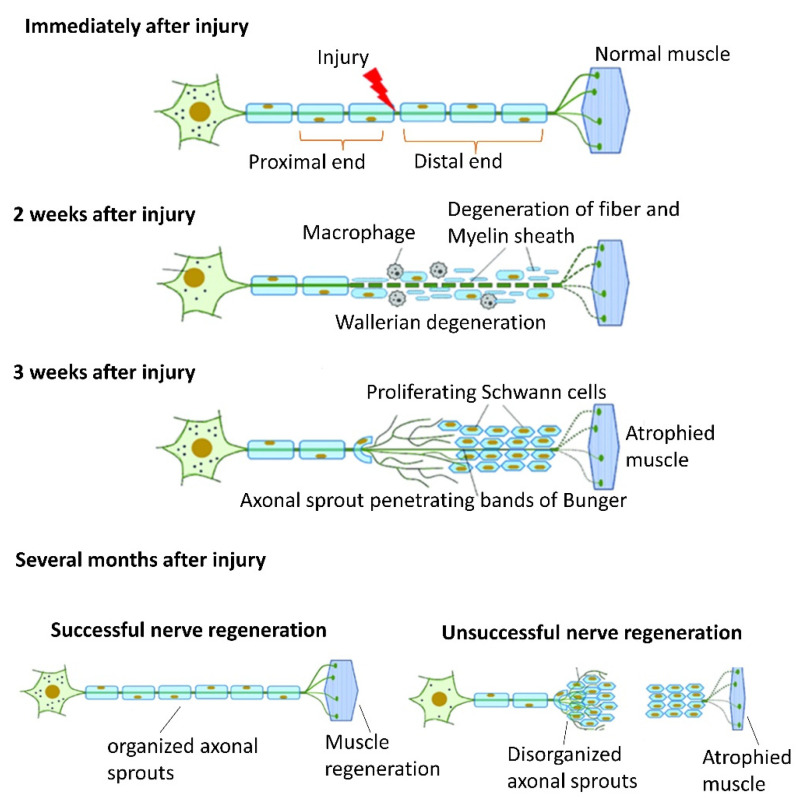
The representative stages of the nerve cell degeneration and regeneration process, reproduced from [29].

**Figure 5 nanomaterials-13-02528-f005:**
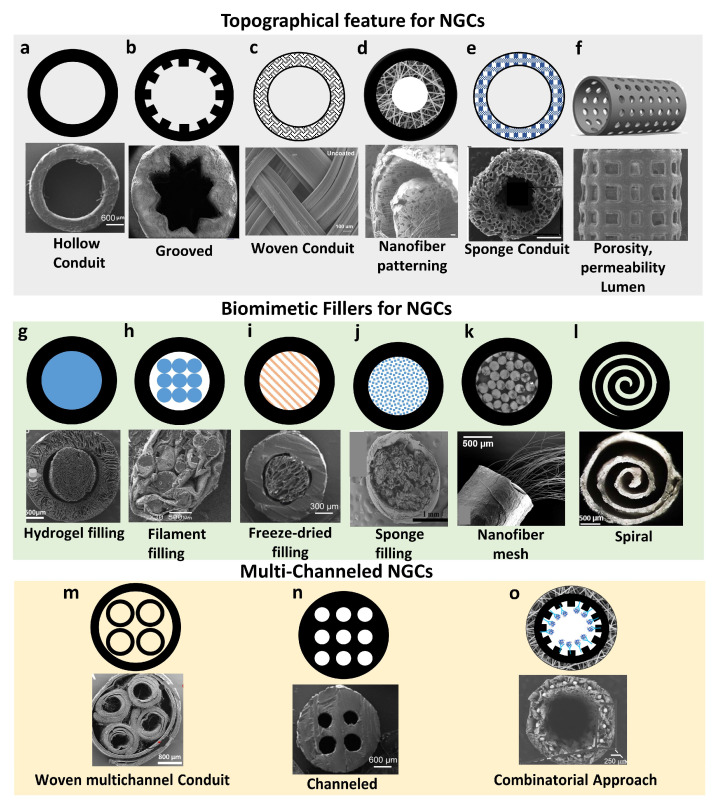
Different NGC architectures. (**a**) Simple hollow conduit of first-generation NGCs [103]. (**b**–**f**) Grooved and porous hollow NGCs [69,101,104,105,106]. (**g**–**l**) Biomimetic fillers for NGCs [105,107,108,109,110,111]. (**m**,**n**) multichannel conduits [101,108]. (**o**) NGC containing multiple biomimetic features [112].

**Figure 6 nanomaterials-13-02528-f006:**
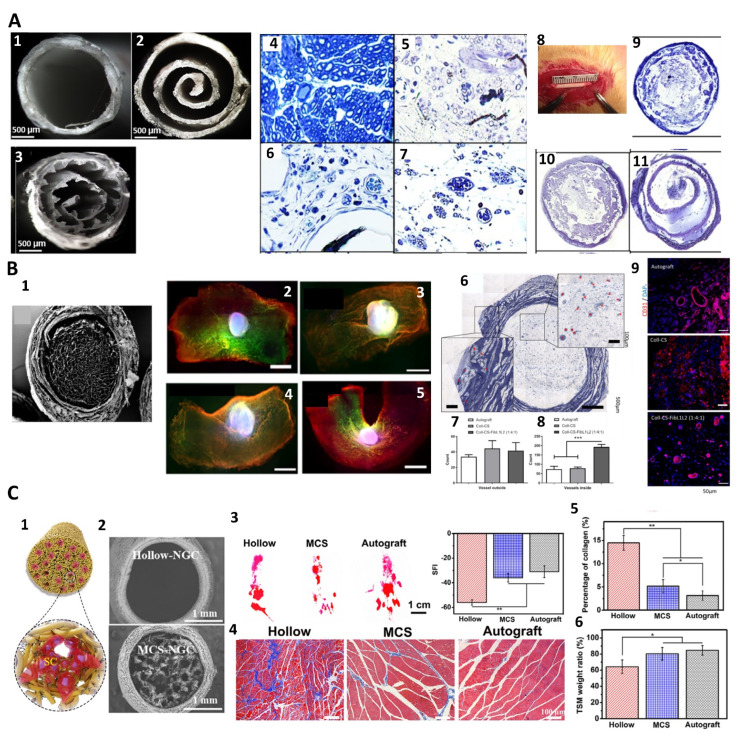
Design considerations for biomimetic NGCs. (**A**) (1) Creation of hollow NGCs via collection of electrospun PCL nanofibers onto a rotating rod; (2) spiral NGC made via solvent casting, salt-leaching, and electrospinning of PCL; (3) spiral microgrooved NGC made via solvent casting and electrospinning of PCL; (4–11) histological staining of cross sections indicates the presence of blood vessels and myelinated axons in a 15-mm sciatic gap rat model 4 weeks after implantation [111]. (**B**) (1) Outer wall of the filled NGC was made via crosslinking type 1 collagen using 0.3% formaldehyde in a mold. The inner filling was made using lyophilizing collagen and chondroitin-6-sulfate containing various biomolecules [49]; (2–5) fluorescence imaging of Schwann cells expressing neuro-regenerative markers; (6) toluidine blue staining of blood vessels, marked with a red asterisk, throughout the NGC; (7–9) analysis of blood vessel infiltration in both implanted conduits and autografts after their respective implantation [49]. (**C**) (1) Illustration of Schwann cells infiltrating multichanneled NGCs made via molding and freeze-drying; (2) SEM images of multichanneled sponge (MCS) and hollow NGC; (3) rat track analysis and sciatic function index (SFI) of NGCs following 12 weeks of implantation; (4,5) Masson’s trichrome staining for collagen 12 weeks post-implantation; (6) triceps sural muscle weight ratio analysis [62].

**Figure 7 nanomaterials-13-02528-f007:**
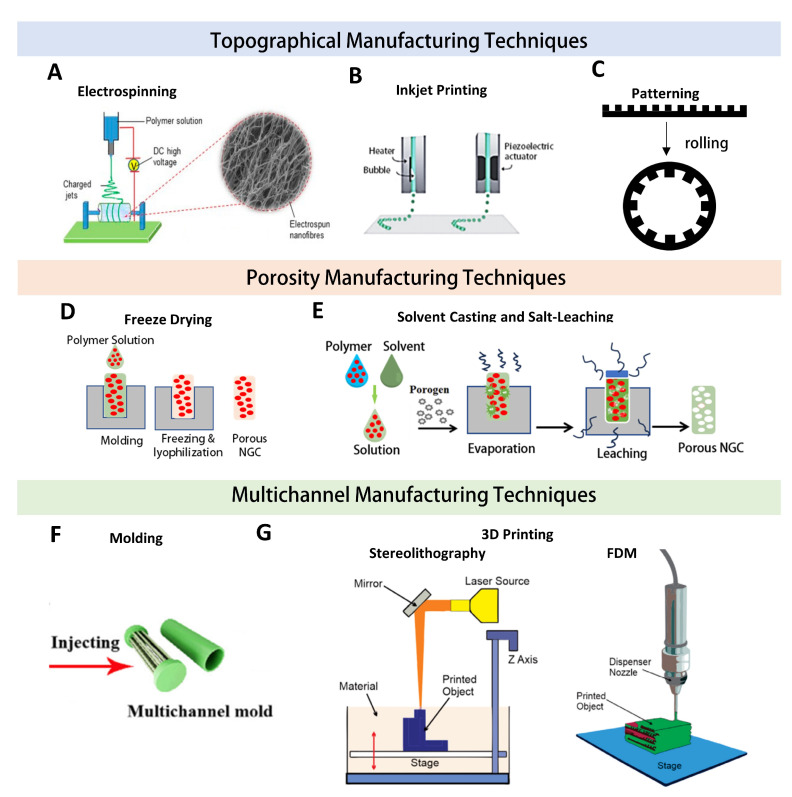
Schematics of various fabrication technologies for generating biomimetic NGCs. Topographical manufacturing techniques include (**A**) electrospinning, (**B**) inkjet printing, and (**C**) patterning. Figures adapted from [8,103]. Porosity manufacturing techniques include (**D**) freeze-drying and (**E**) solvent casting and salt-leaching. Multichannel manufacturing techniques include (**F**) molding and (**G**) 3D printing (stereolithography and fused deposition modeling). Figures adapted from [2,136].

**Table 1 nanomaterials-13-02528-t001:** Representative synthetic and natural materials used for biomimetic NGCs.

**Synthetic Biomaterials**			
**Biomaterials**	**Advantages**	**Disadvantages**	**Ref.**
PLA	High tensile strength Biodegradable	Long degradation time Limited bioactivity Hydrophobic Acidic byproducts may trigger inflammation or an immune response	[14,35,36]
PLGA	High tensile strength Modifiable degradation rate dependent on the lactic and glycolic acid ratio	Limited bioactivity Hydrophobic Acidic byproducts may trigger inflammation or an immune response	[15,37,38]
PLCL	High tensile strength Improved degradation rate compared to PCL	Long degradation time Limited bioactivity Hydrophobic Acidic byproducts may trigger inflammation or an immune response	[40]
PPy	Conductive Easy polymerization	Limited biodegradability Conductivity lost at elevated temperatures	[41,42]
PANI	Conductive High tensile strength Bioactive Biodegradable	Degradation causes a loss of conductivity Less conductive in the acidic conditions found during the initial stages of nerve regeneration	[43,44]
PEDOT: PSS	Conductive Forms hydrogels and films Highly elastic and flexible	Limited long-term stability Sensitive to moisture Limited biodegradability May trigger an immune response	[41,45]
CNTs	Conductive High compressive strength Naturally porous material	Difficult to fabricate Clustered CNTs can hinder cell interactions	[41,42,46]
rGO	Conductive High tensile strength	Difficult to disperse Clustered rGO can hinder cell interactions Potentially cytotoxic May trigger immune responses	[46]
**Natural Biomaterials**			
**Biomaterials**	**Advantages**	**Disadvantages**	**Ref.**
Collagen	Derived from ECM Contains cell adhesion motifs Enhances SC proliferation, migration, and axonal growth Minimal adverse reactions Hydrophilic	Low mechanical strength Rapid biodegradation Highly variable properties dependent on source and extraction process Potentially immunogenic	[45,46,47,48,49,50,51]
Gelatin/GelMA	Derived from collagen Contains cell adhesion motifs Enhances SC proliferation, migration, and axonal growth Hydrophilic	Temperature-sensitive (gelatin) Potential UV toxicity (GelMA) Low mechanical strength Rapid biodegradation Potentially immunogenic	[44,46,52,53]
Fibrin	Derived from ECM Contains cell adhesion motifs Encourages SC attachment and migration Hydrophilic	Little to no mechanical stability Low mechanical strength Rapid biodegradation	[52,54,55]
Hyaluronic acid	Derived from ECM Hydrophilic Has a lubricating and cushioning effect Smooth surfaces promote axonal outgrowth	Low mechanical strength Rapid degradation Lacks inherent guidance cues	[56,57,58,59,60,61]
Silk fibroin	Degradation rate closely matches the nerve regeneration rate High tensile strength Less variable properties compared to ECM-derived biomaterials	Expensive compared to ECM-like materials Lacks inherent guidance cues Potentially immunogenic	[62,63]
Chitosan	Degradation rate closely matches the nerve regeneration rate Less variable properties compared to ECM-derived biomaterials Interacts strongly with laminin, fibronectin, and collagen, which improve cell adhesion Antimicrobial and antifungal properties	Low mechanical strength Limited water solubility Acid soluble—improper neutralization may generate an immune response	[52,63]

PLA: poly(lactic) acid; PLGA: poly (lactic-co-glycolic) acid; PLCL: poly(lactide-co-epsilon-caprolactone); PPy: polypyrrole; PANI: polyaniline; PEDOT:PSS: poly(3,4-ethylenedioxythiophene) polystyrene sulfonate; CNTs: carbon nanotubes; rGO: reduced graphene oxide; GelMA: gelatin–methacryloyl.

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
