# Peer review of "Advances in Biomimetic Nerve Guidance Conduits for Peripheral Nerve Regeneration"

_nanomaterials, 2023, doi:10.3390/nano13182528_

Round 1

Reviewer 1 Report

no comments

English needs moderate editing.

Reviewer 2 Report

This paper provides an interesting review on the biomimetic nerve guidance conduits for peripheral nerve regeneration. This topic is specifically pressing and current and worth a deeper investigation.

The manuscript is relatively well organized and clear. The figures and tables are also informative and understandable. It summarizes well the progress on this research field and gives valuable contribution to the scientific literature.

My minor comment on the paper as follows:

The reference list cannot be considered up-to date, there are only a few recent article cited so it needs to be updated and expanded with relevant and current published papers on this area, detailing the achievements described in them so far.

Reviewer 3 Report

This good work struck me immediately. First, the authors very well chose the topical focus of the review. Undoubtedly, the role of the peripheral nervous system (trophic, coordinating, conducting as part of the afferent and efferent tracts, signaling) is difficult to overestimate. The integrity of the nervous system is an important indicator of health, and the search for effective ways to regenerate the peripheral nerves is one of the most important tasks of regenerative medicine. I really liked that the article was written competently and logically, the style of presentation immediately allows the reader to feel the problem and assess the scale of the problem. I especially want to note that the article is beautifully illustrated and all the drawings are very necessary and informative. Once again I will say that I received intellectual pleasure from reading this manuscript. Below are my comments for the minor revision before publishing:

1) 2 paragraph - the title needs to be bold

2) in the Introduction section, briefly describe the structure of the review that awaits readers

Reviewer 4 Report

"The paper titled 'Advances in Biomimetic Nerve Guidance Conduits for Peripheral Nerve Regeneration' is well-addressed. It provides detailed information about the considerations involved in designing a scaffold for nerve regeneration and the potential materials used. The paper contains valuable information for the scientific community, serving as a guide for future research projects. However, there are some aspects that could be enhanced. Here are some suggestions:

-Figure 3 is unclear in its final part. The image seems to indicate that there is subsequent damage after nerve regeneration. Please consider improving the clarity of this portion of the figure.

-It would be beneficial to provide examples of fillers used in the context of nerve guidance conduits.

-Please review Table 2, particularly in the 'Electrospinning' section. You mention fiber diameter as both an advantage and a disadvantage. Similarly, in the 'Solvent casting and Salt-leaching' section, pore size could be discussed in more detail.

-It would be helpful to provide additional information about the methods employed to manufacture the scaffold in the given examples.

-In the 'Fabrication' section, consider mentioning hybrid methodologies. There have been significant advancements in tissue engineering, including the production of 3D scaffolds using techniques like templates or a combination of electrospinning and 3D printing, etc.

no comments 

Reviewer 5 Report

The paper is devoted for review of advances in biomimetic nerve guidance conduits for peripheral nerve regeneration. The topic is generally interesting, however the paper contain unexplained places (below) and need major revisions.

The aim of the paper should be more clearly formulated.

It should be explained how was obtained the data in Fig. 2. The information in which life situations such nerve damages occurs should be also addressed.

Table 1, it difficult to compare properties of very different materials, like CNT or PLA. It really possible PLA by CNT in biological applications? It would be useful to specify which materials properties (electrical conductivity, mechanical properties, porosity) should be for regeneration damaged nerve.

The sentence at lines 243-244 should be explained.

Conclusions should be rewritten in more informative way.

 Numbers and measurements units should be written separately, for example line 150, it should be 50 mmHg.

Round 2

Reviewer 5 Report

Authors make proper corrections according to

reviewer remarks and I suggest to publish the paper as it is.